# 3D Organoids for Regenerative Endodontics

**DOI:** 10.3390/biom13060900

**Published:** 2023-05-28

**Authors:** Fang-Chi Li, Anil Kishen

**Affiliations:** Dental Research Institute, Faculty of Dentistry, University of Toronto, Toronto, ON M5G 1G6, Canada; fangchi.li@utoronto.ca

**Keywords:** organoids, in vitro models, cell interactions, regenerative endodontics

## Abstract

Apical periodontitis is the inflammation and destruction of periradicular tissues, mediated by microbial factors originating from the infected pulp space. This bacteria-mediated inflammatory disease is known to interfere with root development in immature permanent teeth. Current research on interventions in immature teeth has been dedicated to facilitating the continuation of root development as well as regenerating the dentin–pulp complex, but the fundamental knowledge on the cellular interactions and the role of periapical mediators in apical periodontitis in immature roots that govern the disease process and post-treatment healing is limited. The limitations in 2D monolayer cell culture have a substantial role in the existing limitations of understanding cell-to-cell interactions in the pulpal and periapical tissues. Three-dimensional (3D) tissue constructs with two or more different cell populations are a better physiological representation of in vivo environment. These systems allow the high-throughput testing of multi-cell interactions and can be applied to study the interactions between stem cells and immune cells, including the role of mediators/cytokines in simulated environments. Well-designed 3D models are critical for understanding cellular functions and interactions in disease and healing processes for future therapeutic optimization in regenerative endodontics. This narrative review covers the fundamentals of (1) the disease process of apical periodontitis; (2) the influence and challenges of regeneration in immature roots; (3) the introduction of and crosstalk between mesenchymal stem cells and macrophages; (4) 3D cell culture techniques and their applications for studying cellular interactions in the pulpal and periapical tissues; (5) current investigations on cellular interactions in regenerative endodontics; and, lastly, (6) the dental–pulp organoid developed for regenerative endodontics.

## 1. Introduction

Normal periapical healthy tissues consist of cementum, the periodontal ligament (PDL), and an alveolar bone. Apical periodontitis is the inflammation of the periapical tissues and is mainly mediated by the microbes and their byproducts in the root canal lumen [1,2]. The interaction between the microbial virulence factors or the pathogen-associated molecular pattern (PAMP) and the host immune response disrupts tissue homeostasis, resulting in the inflammation and destruction of periapical tissues [2]. In apical periodontitis, differences in cellular crosstalk and signaling from cell-to-cell and cell-to-extracellular matrix (ECM) interactions influence the expression of inflammatory mediators/cytokines, regulating the periapical host response to microbes. The degradation of the ECM, involving the periapical bone, cementum, and dentin, also contributes to disease progression [2,3]. Apical periodontitis in immature permanent teeth impairs further root development. This incompletely developed immature permanent tooth presents important clinical challenges. Most notably, thin and short roots compromise a tooth’s mechanical integrity, increasing its predisposition to root fracture [4,5]. Typically, the healing of apical periodontitis is considered crucial for the continuation of root development [6].

Regenerative endodontic procedures aimed at treating immature teeth with apical periodontitis have recently gained significant attention. The goal of these treatment procedures is to achieve biologically based periapical wound healing, increase the width and length of the root and possibly restore the neural function of dental pulp [7]. In this respect, understanding the cellular functions and immune cell – host tissue-forming cell crosstalk, as well as the cells’ signaling mechanism, would not only aid in understanding the disease process of apical periodontitis but also aid in developing strategies for tissue regeneration by immune modulation. In this respect, macrophages account for the largest proportion of immune cells in apical periodontitis [8,9,10]. The array of pro-inflammatory and anti-inflammatory mediators produced by macrophages and their interaction with other immune and precursor cells reveal their important role in the development and progression of apical periodontitis [11]. This narrative review covers the fundamental aspects of (a) the pathogenesis of apical periodontitis, (b) the challenges in regenerative approaches for immature teeth, (c) the crosstalk mechanisms between mesenchymal stem cells and macrophages, (d) 3D cell culture techniques and their applications in pulpal and periapical tissue dynamics and (e) current investigations of cellular functions and interactions in regenerative endodontics.

## 2. Apical Periodontitis

Apical periodontitis is the inflammation and destruction of periradicular tissues mediated by microbial virulence factors of endodontic origin. Dental pulp offers the first line of defense against invading microbial threats in a susceptible tooth. Thus, in the initial step of the disease process, the dental pulp becomes inflamed, infected, and necrotic due to autogenous oral microflora. The invasion of pulp space by the microbes and the egress of microbes/byproducts into the periapical region can induce a range of periapical inflammatory responses. The periapical host response activates several classes of cells and an array of intercellular messengers, antibodies, and effector molecules [2] (Figure 1). Intracanal bacteria and their byproducts, including proteins, carbohydrates, and lipids, that form pathogen-associated molecular patterns (PAMPs) and damage-associated molecular patterns (DAMPs) induce immune responses by host cells via the activation of pathogen-recognizing receptors (PRRs) in the host immune cells (e.g., Toll-like receptors) [12]. Among different modulins that induce the formation of cytokine networks and host tissue pathology, lipopolysaccharides (LPS) are a key component. They are the major constituent of the cell wall of Gram-negative bacteria and have been shown to act as endotoxins that elicit a variety of immune responses in odontoblasts, fibroblasts, stem cells associated with dentin–pulp/periodontal tissues, endothelial cells, and macrophages [2,13]. LPSs not only signal the endothelial cells to express adhesion molecules but also activate macrophages to produce several molecular mediators, such as tumor necrosis factor-α (TNF-α) and interleukins (IL) [14]. Via Toll-like receptor 4 (TLR-4), LPSs also activate specific pathways in the host cells, resulting in chemokine/cytokine production [12,14].

Host-derived mediators, which are induced by the infective process, are critical in stimulating periapical inflammation and tissue destruction. These mediators play important roles in combating infection but may do so at the price of promoting tissue damage. Tissue destruction in the periapex involves the resorption of bone and its replacement by granulation and/or cystic transformation, which is extensively infiltrated by leukocytes. Currently, it has been established that the pathogenic effects of pulpal infections on the periapical tissue are predominantly indirect and operating via the stimulation of host-derived soluble mediators such as chemokines/cytokines rather than by the direct necrotizing effects of bacteria on tissue [15].

Generally, the earliest periapical response to inflammation involves the migration of polymorphonuclear neutrophils (PMNs) and monocytes [16,17]. The massive infiltration of neutrophils is characteristic of the acute phases of apical periodontitis. Chemokines such as IL-8, monocyte chemoattractant peptide-1 (MCP-1), and macrophage-derived chemokine (MDC) are present in periapical tissues and are likely to be involved in stimulating periapical monocytes and leukocyte infiltration [18]. The pro-inflammatory cytokines IL-1 and TNF-α are expressed early in response to infection and subsequently induce the production of downstream mediators such as IL-6 and IL-8 [19]. IL-lα/β, TNF-α/β, IL-6, and IL-11 possess varying levels of bone-resorptive activity. IL-1β shows 500-fold more potent bone-resorptive activity than TNFs [20]. In addition, IL-1 and TNF-α contribute to tissue destruction by inducing PGE2 and matrix metalloproteinases [21,22]. On the other hand, IL-2, 4, 5, 6, 10, and 13 are secreted by macrophages and T helper 2 (TH2) during inflammation, suppressing IL-1 production [15,23]. IL-4 and IFN-γ may also inhibit IL-1-stimulated bone resorption [24,25].

Members of the transforming growth factors type beta (TGF-β) superfamily are critical regulators of cell growth, differentiation, repair, and inflammation [26]. During the early phases of inflammation, TGF-β1 acts a chemoattractant for monocytes and lymphocytes, recruiting them to the site of injury, and in the later phases, exerts a potent suppressive effect on the proliferation and differentiation of both T- and B-lymphocytes [27]. Furthermore, TGF-β1 inhibits the production and antagonizes the biological activities of IL-1, IL-2, IL-6, TNF-α, and lFN-γ while inhibiting bone resorption [28]. TGF-β1 is a powerful negative regulator of inflammation, stimulating collagen synthesis, neovascularization, and fibroblast proliferation [29]. Regardless of the formidable defense mechanisms mentioned above, the host immune response is unable to eliminate the microbes that are well established in the morphological complexities of the infected root canal, which is beyond the reaches of the host immune mechanism [2]. Therefore, apical periodontitis, although it may present clinically as symptomatic or asymptomatic, may not be a self-resolving inflammatory process.

## 3. Effects of Apical Periodontitis in Immature Roots

The complete development of the root takes three years after tooth eruption is complete. An immature root is shorter, features a thinner dentin wall, and lacks apical constriction. The apical aspect of the immature root contains the “apical papilla”, which is the congregation of progenitor stem cells, is of ecto-mesenchymal origin, and has the potential to form the dentin–pulp complex [30,31]. In immature teeth with apical periodontitis, the intracanal microbial virulence factors interfere with the process of root development resulting in hindered root maturation. These teeth with immature roots are prone to fracture and subsequently tooth loss [32,33].

An earlier study showed that the apical papilla of immature permanent teeth with deep caries had extremely reduced cellularity, while the Hertwig’s epithelial root sheath (HERS) was discontinuous or absent in cases with pulpitis and pulp necrosis [34]. It is known that young patients have a stronger host immune mechanism than older patients. This is attributed to the efficient blood circulation at the open root apex, allowing the transportation of the cellular and molecular components of innate and adaptive immune responses to the canal space [35]. Clinical investigations have revealed that the apical papilla was able to survive the process of pulp necrosis [36,37]. This was attributed to the apical location of the apical papilla, which benefits from collateral circulation, and/or endothelial trans-differentiation, which induces angiogenesis during inflammation [31,37]. It appears that stem cells from the apical papilla (SCAP) and surrounding stem cells are equipped to receive nutrients and oxygen via diffusion from the surrounding apical tissues for survival and for maintaining their differentiation potential in adverse conditions, such as apical periodontitis and abscesses. Therefore, the current regime of treatment has shifted from conventional root canal treatment and apexification to regenerative endodontic procedures aimed at promoting root development in immature permanent teeth with pulpal necrosis and apical periodontitis [38].

## 4. Challenges in Regenerative Endodontic Procedures

Regenerative endodontic procedures (REPs) are defined as biologically based procedures designed to replace damaged structures, including dentin and roots, as well as cells of the dentin–pulp complex [39]. REPs have emerged as viable alternatives for the treatment of immature teeth with pulp necrosis [38]. REPs rely heavily on the chemical debridement of the root canal following minimal or no instrumentation. The root canal system is further disinfected with an intracanal medication, followed by the induction of bleeding into the canal from apical tissues. Tricalcium-silicate-based materials are used as intracanal barriers and are followed by bonded composite resin restoration [40]. Currently, three treatment outcomes are anticipated for REPs: (i) the resolution of clinical signs and symptoms (the healing of apical periodontitis); (ii) further root maturation (increases in root width and length); and (iii) the return of neurogenesis (the restoration of pulpal nerve function) [40].

Current treatment standards and position statements have recommended REPs as a treatment alternative to traditional apexification for immature permanent teeth with necrotic pulps [38]. Clinical studies and case reports have demonstrated acceptable clinical outcomes with REPs on immature permanent teeth with apical periodontitis [41,42,43,44]. However, an in vivo histological analysis has demonstrated the formation of fibrous connective tissues and cementum-/bone-like tissues in the root canal space rather than the regeneration of the dentin–pulp complex [45,46]. These findings indicate that repair could take place following REPs in immature teeth instead of the true regeneration of the tissues involved. The histological studies conducted on clinical samples have emphasized (i) the need to use disinfection protocols that create an environment that is better suited for an organized pulp-like tissue formation and (ii) the application of more advanced bioengineering approaches such as cell-based therapy using stem cells in a scaffold/growth factor construct and/or the use of chemokine-based cell homing (cell-free) approaches [40].

A cell-based method that relies on delivering stem cells into the root canal space has been shown to regenerate the dentin–pulp complex [47,48]. However, it is challenging to apply the therapeutic armamentarium in routine clinical practice due to the cost related to stem cell extraction enrichment and therapeutic techniques. The use of chemotaxis-based cell homing (cell-free method) therapies has the advantage of not requiring the laceration of the apical papilla to evoked bleeding, minimizing the disruption between apical papilla and the HERS as a key determining factor that guides root development [49]. This method used in previous studies resulted in the formation of pulp-like tissues; however, it did not result in consistent mineralization [49,50]. A major concern in the cell-homing technique for REPs is that the source of the stem cells that leads to the formation of neo-tissues does not belong to the dentin–pulp complex.

## 5. Mesenchymal Stem Cells (MSCs) and Stem Cells from Apical Papilla (SCAP)

Mesenchymal stem cells (MSCs) represent a class of cells from human and mammalian bone marrow and periosteum that is different from hematopoietic stem cells [51,52]. They possess an extensive proliferative potential and an ability to differentiate into a variety of mesodermal phenotypes and tissues, including osteocytes, adipocytes, chondrocytes, myocytes, cardiomyocytes, and neurons [53]. The surface markers, growth, function, and differential potential of MSCs have been well reviewed in detail in previous studies [51,52]. Apart from the bone marrow, MSCs are also located in other tissues of the human body, and most stem cells found in the orofacial region are MSCs [54]. The stem cells surrounding the periapical region are the most likely to be involved in regenerative endodontic procedures (REPs). These include stem cells from the apical papilla (SCAP), periodontal ligament stem cells (PDLSCs), bone marrow stem cells (BMSCs), inflamed periapical progenitor cells (iPAPCs), and dental pulp stem cells (DPSCs) (if vital pulp tissue is present apically).

Dental papilla, derived from ectomesenchyme, evolves into the dentin–pulp complex [30]. The apical papilla is the apical portion of the dental papilla during root development, located precisely apical to the epithelial diaphragm, with a cell rich zone separating it from the dental pulp [55]. The stem cells from the apical papilla (SCAP) hold the ability for odontogenic differentiation, forming the dentin–pulp complex. They have the potential to differentiate into odontoblasts, osteoblasts, neurocytes, and endothelial cells [55,56]. During tooth development, the HERS secretes laminin 5 to induce dental papilla cells to migrate, attach, grow, and differentiate [57], while TGF-β induces dental papilla cells to differentiate into odontoblasts [58]. The migration and differentiation of SCAPs during root development are regulated by the HERS via a series of complex epithelial–mesenchymal interactions. Similarly to other mesenchymal stem cells, SCAPs express STRO-1 and CD146, which are recognized as early MSC markers [56]. SCAPs are also characterized by the expression of surface and intracellular molecules: they express pluripotent markers such as octamer-binding transcription factor-3/4, sex-determining region Y-box 2, a nanog homeobox, CD13, CD24, CD29, CD44, CD49, CD51, CD56, CD61, CD73, CD90, CD105, CD106, CD166, NOTCH3, and vimentin [59,60]. SCAPs express MSC-associated markers and are capable of self-renewal, proliferation, and multilineage differentiation (odontogenic, osteogenic, neurogenic, angiogenic, and weak adipogenic potential) [55,61]. During pulp necrosis and apical periodontitis, even though the viability of SCAP may not be significantly affected according to clinical observation and in vitro research, previous studies have shown that the dentinogenic potential of SCAPs was inhibited when exposed to biofilm/LPSs [37,62,63] and pro-inflammatory cytokines (TNF-α and IL-1-β) [64]. Furthermore, their osteogenic potential may be upregulated, leading to bone-like tissue formation rather than dentin tissue regeneration [37,62].

## 6. Macrophages

Macrophages (MQs) are derived from bone marrow stem cells in response to monocyte colony-stimulating factor to form monocytes (the precursors of MQs), which circulate in the blood [65]. After the initiation of inflammation, they migrate to inflammatory tissues and mature into MQs, maintaining tissue homeostasis and serving as the first line of innate immunity [65]. MQs are professional phagocytic cells that internalize and kill bacteria via several mechanisms, some of which are part of innate immunity, while others require the presence of specific antibodies against the bacterium and should be considered part of the effector arm of specific, acquired immunity [11]. Macrophages, lymphocytes, and plasma cells have been consistently presented as inflammatory infiltrates in both periapical granulomas and cysts [10,66]. In the periapical inflammatory lesions, MQs have been shown to be the predominant immunocompetent cells throughout the development of the apical lesion [17]. They occupy up to 46% of the periapical inflammatory cells and outnumber T lymphocytes in human periapical granuloma [8,9]. MQs increase in numbers during the first 10 days, maintain this level through day 60, and decline gradually thereafter [16]. The plastic characteristics of MQ elicit its polarization, by which MQs differentiate into specific phenotypes (M1/M2) and have specific biological functions in response to microenvironmental stimuli [67,68,69]. M1 macrophages present antibacterial, antigen-presenting functions and facilitate T helper type 1 (Th1) response, while M2 macrophages suppress Th1 and promote Th2, angiogenesis, and wound healing. Therefore, MQs polarize to M1 due to the stimulations of intracellular pathogens, bacterial cell wall components, and hallmark Th1 cytokines such as IFN-γ [70]. Additionally, M2 would be stimulated by IL-4 and IL-13(M2a), immune complexes, Toll-like receptors (TLRs) or apoptotic cells (M2b), glucocorticoids, and TGF-β or IL-10 (M2c) [71].

Due to the plasticity of MQs in response to the environmental stimuli, different surface markers are expressed: CD68 is generally presented on MQs; CD 80 and 86 are more associated with M1; CD 163, 204, and 206 are more related to M2 [72]. During inflammation, M1 macrophages produce IL-1β, IL-6, IL-12, IL-23, TNF-α, ROS, and proteolytic enzymes [73,74] and release CXCL9, CXCL10, and CXCL13 to attract Th1 lymphocytes [68,75], whereas M2 macrophages produce high levels of anti-inflammatory cytokines IL-10, CCL17, CCL22, and CCL24 to recruit Th2 cells, basophils, and mast cells, thereby promoting Th2 responses as well as PDGF, VEGF, and EGF to promote angiogenesis and wound healing [75,76,77]. It was observed in a previous clinical study that MQs exhibited a polarization switch towards M1 in an apical lesion and symptomatic apical periodontitis and exhibited a reduced M2 differentiation profile based on a reduction in CD163 expression levels in symptomatic over asymptomatic apical periodontitis [78]. Moreover, MQ polarization might direct the development of apical periodontitis towards apical granulomas or radicular cysts. One clinical study presented that radicular cysts are characterized by the M1 polarization of MQs, while apical granulomas show a significantly higher degree of M2 polarization [79]. Since immunomodulation plays a crucial role in wound healing, Hussein and Kishen comprehensively reviewed the immunomodulatory effect of antimicrobial intracanal medications applied in endodontics with specific emphasis on antimicrobial nanomaterial-based approaches [80]. It is suggested that the desirable immunomodulatory materials may shift from M1 to M2, creating a pro-regenerative and anti-inflammatory environment [81]. Thus, it would be valuable to investigate the role of MQ polarization in periapical inflammation and wound healing in immature roots.

## 7. Mesenchymal Stem Cell–Macrophage Crosstalk

The repair and regeneration of wound healing is guided by the delicate interaction/crosstalk between tissue-forming progenitor cells and immune cells during inflammation [72,82]. This crosstalk between stem cells and immune cells modulates both innate and adaptive immune reactions via juxtacrine and paracrine signaling [69,83]. Studies have demonstrated that MQs affect the migration, proliferation, and survival of mesenchymal stem cells (MSCs). It was also demonstrated that M1 type MQs may inhibit the growth of human MSCs in vitro and induce the apoptosis of MSCs, while M2 type MQs promote MSC proliferation and migration [84,85]. In the healing of bone, the initial inflammatory reaction as well as the pro-inflammatory MQ activation contributes to the recruitment of MSCs of osteoprogenitor and vascular progenitor cells to the wound. The signals that control progenitor cell homing include the chemokines CCL2, CXCL8, and SDF-1, all of which are secreted by activated MQs [86,87]. Guihard et al. found that conditioned media from human monocytes stimulated with LPS or TLR ligands enhanced bone formation via human bone marrow MSCs [88]. Enhanced osteogenesis was also observed by several groups, who demonstrated that M1 MQs promote osteogenesis in MSCs via the COX-2-PGE2 pathway [88,89,90]. MQs have crucial influence on the differentiation of MSCs. In addition, MSCs express more immune regulatory genes after being co-cultured with MQs. They stimulate the release of IL-10 by MQs via a PGE2-dependent pathway after LPS stimulation. This production of PGE2 by MSCs is activated by the necessary secretion of TNF-α and iNOS by MQs [91]. Therefore, it is suggested that pro-inflammatory cytokines produced by MQs stimulate MSCs to produce PGE2 and IL-1RA, among other immune modulators [92].

MSCs repair damaged tissues by responding to inflammation. They migrate to injured sites and influence the microenvironment by releasing molecules in the environment, promoting the reparative or regeneration process in wound healing [93]. The combination of INF-γ with another pro-inflammatory cytokine (TNF-α, IL-1α, or IL-1β) further activates MSCs in damaged or inflamed tissues [94]. It is indicated that the immunomodulatory properties of MSCs depend largely on the expression of soluble factors. It is important to recognize that cell-to-cell contact is also an important functional mechanism [95]. At present, there is not much information available on the juxtacrine signaling between MSCs and immune cells. In a co-culture model, MSCs suppressed the LPS-induced production of pro-inflammatory cytokines (TNF-α, IL-1β, and IL-6) and increased the secretion of IL-10 by murine MQs [91,96]. The MSCs induced MQs to adopt an enhanced regulatory phenotype via increasing IL-10 and reducing TNF-α and IL-12 secretion predominantly via prostaglandin E2 (PGE2) synthesis [91,97]. MSC-derived soluble factors such as IL-10, PGE2, and IL-1β are key molecules involved in the crosstalk between MSCs and MQs, particularly for shifting polarization from the M1 to the M2 phenotype [98]. MSCs also regulate MQ chemotaxis by producing CCL2 and CCL4 to attract monocytes and MQs. Another important molecule associated with the crosstalk between MSCs and MQs is transforming growth factor β (TGF-β), which plays a key role in the immunosuppressive function of MSCs [99]. TGF-β secreted by MSCs promotes the M2 polarization of MQs and regulates the inflammatory response mediated by MQs to improve pro-inflammatory conditions [100]. The above studies clearly highlighted that the chemokine- and/or cytokine-mediated crosstalk between tissue-forming stem cells and MQs via paracrine as well as juxtacrine interactions has a critical role in guiding tissue repair and regeneration.

As previously mentioned, MSCs may be influenced by MQ polarization, while polarized MQs may guide MSC recruitment and differentiation [82]. The crosstalk between MQs and stem cells in immunomodulation potentially regulates the process of tissue regeneration, including wound healing [72]. Therefore, a comprehensive understanding of the interaction/crosstalk mechanisms between MQs and tissue-forming stem/progenitor cells at the periapical region of a tooth is critical for developing therapeutic approaches in regenerative endodontics. The research conducted by our group showed that the presence of SCAPs or MQs influenced the cytokine profile in the inflammatory environments and the differentiation/polarization of the SCAPs and MQs (when comparing the co-culture with each mono cell culture) (data not published yet). This cellular level crosstalk in the periapical region highlighted the importance of understanding the cell-signaling mechanism between mesenchymal stem cells and immune cells and the role of cytokines in periapical wound healing and regenerative endodontics.

## 8. Three-Dimensional Cell Culture Models

In vitro cell cultures are used to study and relate the mechanisms underlying cell behavior in vivo [101]. The majority of traditional cell-based assays have employed two-dimensional (2D) monolayer cells cultured on flat and rigid surfaces as the gold standard. These 2D cell cultures do not adequately consider the natural 3D environment of cells. In an in vivo environment, cells are bordered by adjacent cells and circulating molecules and supported within an ECM in a three-dimensional (3D) manner [101,102]. In order to translate the results from laboratory-based cell culture experiments to in vivo animal or clinical studies, the environments created for laboratory-based in vitro experiments need to simulate the tissue topography and spatial organization, allowing normal cell–matrix and cell–cell interactions. Three-dimensional cell culture models facilitate paracrine and cell-to-cell (juxtacrine) interactions and cell–matrix mediated signaling, which are all crucial to simulate cellular functions in vivo. Tests conducted using 2D cell cultures sometimes provide misleading and unpredictable data for in vivo translation [103]. Previous studies have highlighted the advantages and differences in cell morphology, cell behavior, gene expression, and drug sensitivity between 2D and 3D cell cultures [104,105,106]. The major comparisons between 2D and 3D cell cultures are listed in Table 1. Studies associated with host tissue cells or stem cells in dentoalveolar structures have also found differences between 2D and 3D cell culture methods (Table 2). Some have focused on investigating cell culturing methods for further regeneration application [107,108,109,110,111], while some have aimed at understanding cell behavior in certain environments between 2D and 3D cell cultures [112,113,114]. Additionally, 3D cell culture models have also been developed and designed for studying dental pulp regeneration [115,116] and periodontal regeneration [117] with mostly a solo cell type.

Three-dimensional in vitro models, such as those of cellular spheroids, consist of cell aggregates displaying complex cell–cell and cell–matrix interactions that replicate the natural micro-environment of cells, including natural gradients in the distribution of nutrients, gases, growth factors, and signaling molecules. Different techniques used to culture cells into 3D spheroids are categorized based on the inclusion nature of an ECM matrix: matrix-free or matrix-based [104,118]. The matrix-free technique allows the spheroid to be formed in suspension, turning it into a more solid structure over a period of time via cell-to-cell interactions [119]; the matrix-based technique starts with cells embedded in a hydrogel matrix, such as collagen, Matrigel, or alginate, which serves as the scaffold and provides the shape of the construct formed [120]. This method has the advantage of high control over the type/density of cells, the source/type of the ECM, and the size/shape of the tissue construct resulting in more suitable structures to study disease processes and for drug discovery [118,121]. Many different techniques have been developed to form a 3D cellular structure associated with matrix-based methods, involving different scaffolds and building up the construct with cell sheets and microfluidic devices to mimic the functions of organs [122].

Three-dimensional cell culture also presents some disadvantages and limitations [123]. Organic matrices, which form the ECM, have an additional cost and require more delicate handling skills [124]. Fully embedded structures do not permit access to the basal areas. Thick 3D cultures may have difficulty distributing oxygen/essential nutrients to where they are needed, and collecting cells or secreted factors for biochemical assays is also more challenging in certain 3D models [125]. Thick structures may be challenging in assessing under microscopy: dense matrix/cell structures may be relatively opaque due to the light scattering in ECM gels; large embedded structures (thick) may be difficult to image, limited by the working distance of the objective lens; 3D structures take much more time to capture the “zone of interest” because of the thickness of the constructs [126,127]. Moreover, many 3D culture techniques are cumbersome and time-consuming (low throughput), rendering them unsuitable for drug development screening and research [125]. Even though one of the emerging matrix-based techniques of forming spheroids using micro-fabricated molds to cast bioinks with a high cell-to-ECM composition ratio may overcome some of the limitations mentioned above, these techniques are usually still time-consuming. They are limited in cell type and to low cell densities and show no or limited control over positioning different types of cells in a 3D structure [118]. A universal technique capable of forming tissue constructs of various 3D shapes from a wide variety of cell types at physiologically relevant cell densities and with the ability to precisely position and integrate different cell types in close proximity to each other would be useful to overcome these limitations and provide a more relevant 3D model. These limitations can be circumvented by developing a rapid biofabrication technique using self-assembled collagen-based heterogeneous 3D tissue constructs [118]. This technique has been applied to study periapical tissue responses in our group [128,129]. It is developed by combining the matrix- and cell-directed collagen self-assembly process to form tissue constructs of various 3D shapes. This technique was found to be precise while incorporating different cell types in close proximity to each other within the 3D construct [118]. It can be employed to a wide variety of cell types at physiologically relevant cell densities.

**Table 1 biomolecules-13-00900-t001:** Summary of major differences between 2D and 3D cell culture methods. Information gathered from studies found in References [104,122].

Cellular Characteristics	2D	3D
Exposure to medium/drug	Cells were equally exposed to nutrients/GF that were distributed in medium.	Nutrients/GF/drugs may not be able to fully penetrate the spheroid and reach cells near the core.
Morphology	Sheet-like and flat (stretched in monolayer)	Natural shape
Proliferation	Faster than in vivo	May be faster/slower compared to 2D-cultured cells depending on cell type/3D model system
Gene expression	Often display different gene/protein expression levels compared to in vivo	Often exhibit gene/protein expression profiles that are more similar to in vivo
Cell interaction	Paracrine/juxtacrine	Paracrine/juxtacrine/cell–matrix
Migration	Cells were attached on only 1 side, and less signaling was identified.	Cells were attached on all sides. More obstacles for migration but may be faster. Alteration of mechanism (more signaling).
Stage of cell cycle	Likely same stage (equally exposed to medium)	Spheroids contain proliferating, quiescent, hypoxic, and necrotic cells.
Drug sensitivity	Cells often succumbed to treatment/drug appeared to be very effective.	Cells often are more resistant to treatment and are better predictors of in vivo drug response

**Table 2 biomolecules-13-00900-t002:** Summary of major differences between 2D and 3D cell culture methods associated with the studies of cells in dentoalveolar tissues.

Article	Cell	Aim	3D (Compared to 2D)
Riccio et al. [107]	Dental pulp stem cells (DPSC)	To characterize the in vitro osteogenic differentiation of DPSCs in 2D cultures and 3D biomaterials	In Matrigel™, DPSCs differentiated with osteoblast/osteocyte characteristics and were connected by gap junction and, therefore, formed calcified nodules with a 3D intercellular network. DPSCs are able to differentiate in osteogenic lineage both in 2D and 3D surfaces, creating osteoblast-like cells that express specific osteogenic markers and produce mineralized ECM.
Yamamoto et al. [108]	Mouse dental papilla cell (MDP)	To evaluate the effects of 3D spheroid culture on the phenotype of MDPs	3D spheroid culture promotes odonto/osteoblastic differentiation (ALP, DSPP, DMP-1, and mineralized module formation) of MDPs compared to 2D, which may be mediated by integrin signaling.
Kawashima et al. [112]	Dental pulp mesenchymal stem cell (DPMSC)	To investigate the properties of DPMSCs cultured with different methods	Higher expression of odonto-/osteoblastic markers, including ALP, osteocalcin, and DSPP. Mineralized nodules rapidly formed in 3D spheroid cultured DPMSCs compared with 2D monolayer cultured DPMSCs.
Zhang et al. [116]	Dental pulp cell (DPC)	To compare the multilineage potential and extracellular matrix production of hDPC between conventional monolayer cultures and cellular spheroid cultures	Microarray analysis identified 405 genes and 279 genes with twofold or greater differential expression in 3D culture. Gene ontology analysis revealed upregulation of extracellular-matrix-related genes and downregulation of cell-growth-related genes. RT-qPCR analysis showed higher expression levels of osteocalcin, DSPP, and ALP. TEM revealed the morphological characteristics of the fibrillary collagen-rich matrix and cell–cell interactions.
Kim et al. [109]	Dental-follicle-derived mesenchymal stem cells (DFSC)	To analyze the stemness and in vitro osteogenic differentiation potential of 3D spheroid dental hDFSCs compared with conventional monolayer cultured MSCs	Three-dimensional hDFSCs have a higher proportion of cell cycle arrest and a larger number of apoptotic cells. Substantially increased levels of pluripotency markers and ECM protein expression. Notable enhancement in the osteogenic induction potential of spheroids compared to 2D, although no differences were observed with respect to in vitro adipogenesis.
Xu et al. [110]	Dental pulp stem cell (DPSC)	To compare the preparation methods and preliminary mechanisms of differentiation of hDPSCs into insulin-producing cells (IPCs) under 2D or 3D culture conditions	The results of RT-qPCR showed that the mRNA expression levels of islet cell marker genes Pdx1 and Ngn3 and insulin in the 3D group were significantly higher than those in the 2D group. Three-dimensional culture promoted the differentiation of hDPSCs into IPCs at the gene and protein levels. IPCs induced by 3D group are more sensitive to and mature via glucose response.
Bu et al. [111]	Dental pulp stem cell (DPSC)	To evaluate cell morphology, cell viability, and the osteo-, adipo-, and chondrogenic differentiation potential of DPSCs cultured in 3D culture plates and 2D monolayer plate	DPSCs cultured in microsphere-forming plates (3D) presented superior multilineage differentiation capacities and demonstrated higher differentially expressed gene expression in regeneration-related gene categories compared to DPSCs cultured in a conventional monolayer plate.
Jeong et al. [113]	Periodontal ligament stem cells (PDLSC)	To compare the characteristics of PDLSCs cultured using 3D versus conventional 2D methods	The viability of the 3D-cultured cells was decreased, but they showed superior osteogenic differentiation compared to 2D-cultured cells. Different gene expression profiles.
Banavar et al. [114]	Periodontal-ligament-derived mesenchymal stem cells (PDLSCs)	To compare the effects of LPSs on PDLSCs in monolayer and 3D culture	Three-dimensional clumps formed by PDLSCs are more resistant to the effects of LPSs and retain osteogenic potential than cells grown in monolayer cultures, with increased IL-6 secretion suggesting a positive influence on osteogenic induction.

ALP: alkaline phosphate; DSPP: dentin sialophosphoprotein; DMP-1: dentine matrix protein-1.

## 9. Current Investigations on Cellular Interactions in Regenerative Endodontics

Multipotent/pluripotent stem cells surrounding the periapical region have pivotal roles in REPs relying on cell-free methods for the treatment of immature permanent teeth with apical periodontitis. In this regard, SCAPs, periodontal ligament stem cells (PDLSCs), bone marrow stem cells (BMSCs), inflamed periapical progenitor cells (iPAPCs), and dental pulp stem cells (DPSCs) (if vital pulp is still present apically) are some of the important groups of stem cells. From these groups of stem cells, SCAPs and DPSCs are able to differentiate into the cell types, which form the dentin–pulp complex [130], while other groups of stem cells hold the innate ability to form fibrous connective tissue, cementum, and bone, which, as mentioned earlier, is histologically not considered true regeneration. Therefore, many current investigations have been focusing on the viability, proliferation, migration, inflammatory response, and differentiation potential of DPSCs and especially SCAPs in inflammatory environments, which are associated with apical periodontitis or materials used during REPs.

The nature of inflammation and characteristics of immune cells would determine the differentiation potential as well as the matrix deposition attributes of stem cells. Until now, most previous studies experimented with SCAP or DPSC in a solo environment, without studying the interaction between these stem cells and other cell types that coexist in the environment during the disease and healing processes [37,62,63,64,131,132,133]. The simplified in vitro setting is straightforward, easy to reproduce, and useful for characterizing cell behavior in a controlled environment. These mono-cell-culture-based investigations are also useful for studying cell responses to different stimulants. However, understanding the interaction between tissue-forming stem cells and other types of cells, particularly the immune cells, is key to gain a comprehensive understanding of cell functions in a manner that is more realistic to an in vivo environment. It has been shown that the role of host immune cells influences stem cells’ behavior, while the interaction between tissue-forming stem cells and immune cells guides the disease and healing processes. The nature of this interaction would determine the outcome of tissue healing–repair, regeneration, and fibrosis [72,82]. Therefore, many studies engaged co-culture models to understand the cell interactions between DPSC/SCAP and immune cells (details listed in Table 3). The biological properties, differentiation potential, and immunogenic properties of stem cells were highlighted rather than the reciprocal interactions between immune cells and stem cells, which in turn may affect the reparative or regeneration potential. Recent studies from our group demonstrated that the cytokine profile and the pattern of the biomineralization of SCAP differentiation were significantly influenced by the presence of MQs and the simulated pro-inflammatory and anti-inflammatory environments [128,129].

Table 3 summarizes the findings from previous investigations on cellular interactions between DPSC/SCAP and other cell types. The summary provides an overview of the cell culture methodologies employed, the tested cell types, the environments provided, and the experimental outcomes. The review of the literature highlights that the 2D co-cultures have been widely used for studying interactions between cell populations in most of the studies [134]. Direct co-cultures (direct contact; one cell type directly on the other) facilitate physical contact between the different cell types, which allows communication via their surface receptors and gap junctions, defined as juxtacrine communication. There are two common methods for indirect co-cultures (contact-free): (a) method one is to incorporate a physical barrier, such as a semi-permeable membrane, between the cell types in a transwell system; (b) method two is to use a conditioned medium that is first used for culturing one cell type and transferring it to the second cell type [135]. This contact-free method only enables cell signaling via the cell secretome, which is defined as paracrine communication. Even though the 2D co-culture system is easily replicated and easily interpretable with capabilities for long-term culturing to studies permitting juxtacrine/paracrine cell interactions, the physiological relevance of this model and environment is questionable [104].

Several reasons have been suggested to explain the differences between cellular functions in 2D and 3D cultures. Cells in 2D cultures are usually flatter and more stretched than how they would appear in vivo. The atypical cell morphology and additional stress on cells influence many cellular processes including cell proliferation, differentiation, and apoptosis, in addition to gene and protein expression [125,136]. The expression and binding efficiency of the surface receptors to certain stimuli on cells may be dissimilar in 2D cultures, particularly with respect to their structure, localization, and spatial arrangement on cell surfaces [137]. In 3D environments, cells are usually at different metabolic stages, whereas cells in 2D cultures are mainly proliferating cells since the necrotic cells are detached from the surface and easily removed during medium change [104,136]. The oxygen and nutrient gradient in the 3D environment is not exhibited in a 2D cell culture [138]. It was emphasized beforehand that tissue functions are regulated by both cell-to-cell and cell-to-matrix interactions. In addition, the reciprocal communication between the macromolecules of the extracellular matrix (ECM) and the cells forms the basis of ontogenesis, wound healing, and tissue homeostasis [139]. On the other hand, virtually all disease processes are related to alterations in ECM structure, disturbances in ECM metabolism, and/or dysregulation in ECM–cell signaling [140,141]. Hence, it is essential to utilize matrix-based 3D cell culture models that incorporate multiple cell types that exist in an in vivo environment and are involved in periapical inflammation and healing.

The currently developed novel 3D tissue graft model combines SCAPs and MQs in a self-assembled collagen-based construct to study periapical inflammation and healing in immature teeth [118,128,129] (Figure 2). The result from this binary cells tissue construct model [129] revealed different cytokine profiles compared to the result from a 2D transwell co-culture model [142] and, more importantly, formed a self-organized cap-shaped structure simulating the apical papilla in vivo—an apical papilla organoid (Figure 3). The findings highlighted the potential of this novel 3D tissue construct model to study periapical mediators in disease, inflammation, and repair for applications in regenerative endodontics. It has the potential to be further utilized as an optimal model for developing and assessing new treatment strategies/biomaterials in endodontics.

**Table 3 biomolecules-13-00900-t003:** Listed articles that have investigated cell interactions involving DPSCs and SCAPs.

Article	Cells Interaction	Cell Culture	Highlight
Ding et al. [143]	SCAP–PBMC	2D Direct contact co-culture and contact-free co-culture in transwell chamber	SCAPs were weakly immunogenic and suppressed T cell proliferation in vitro via an apoptosis-independent mechanism.
Tang and Ding [144]	DPSC–PBMC	2D Direct contact co-culture and contact-free co-culture in transwell chamber	DPSCs failed to stimulate allogeneic T-cell proliferation.
Dissanayaka et al. [145]	DPSC–endothelial cell	2D Direct contact co-culture Direct contact co-culture on Matrigel-coated well	DPSC:EC co-cultures revealed greater ALP activity compared with cultures with DPSCs alone. The expression levels of ALP, BSP, and DSPP genes further confirmed the greater osteo-/odontogenic differentiation in co-cultures. Matrigel assay showed that the addition of DPSCs stabilized preexisting vessel-like structures formed by ECs and increased the longevity of them.
Yuan et al. [146]	SCAP–endothelial cell	2D Direct contact co-culture on Matrigel-coated well	When EC and SCAPs were co-cultured under hypoxia, the most extensive lattice of vessel-like structures among all groups was detected.
Yuan et al. [147]	SCAP–endothelial cell	2D Direct contact co-culture on Matrigel-coated well Contact-free co-culture in transwell chamber	Co-culture of SCAPs and EC accelerated the formation of vascular-like structures. Enhanced migration of HUVECs by SCAPs could be inhibited by ephrinB2-Fc.
Lee et al. [148]	DPSC–THP-1 MQ	2D Direct contact co-culture DPSCs directly on MQs	DPSCs/I-DPSCs suppressed TNF-α secretion, but not IL-1β secretion, by MQs. Immunosuppressive effect of DPSCs/I-DPSCs on MQs was mediated by IDO activity.
Omi et al. [149]	DPSC–RAW MQ	2D DPSC-conditioned media on MQ	DPSC-conditioned media increased gene expressions of M2 markers in RAW264.7 cells. DPSC-conditioned media showed no effect on the proliferation/viability of RAW264.7 cells.
Jin and Kim [150]	DPSC–endothelial cell	3D DPSC cultured in porous microcarrier co-cultured with EC in hydrogel	Genes related with osteogenesis and angiogenesis were significantly upregulated by the co-cultures with respect to the mono-cultures. Future application for bone tissue engineering
De Berdt et al. [151]	SCAP–BV-2 microglial cell SCAP–SCOS SCAP–SCOS–OPC	2D Direct contact co-culture Direct contact co-culture Direct contact tri-culture	Co-culture of SCAPs with LPS-treated BV-2 cells induced a significant decrease in TNF-α mRNA expression and increase in arginase-1 mRNA compared to BV-2 mono-culture. Co-culture of SCOS with SCAPs showed an increase in staining for mature oligodendrocyte markers in SCOS compared to SCOS alone and a significant increase in activin-A gene expression. SCOS–SCAP promotes adult OPC differentiation.
Whiting et al. [152]	DPSC–PBMC DPSC–NKC SCAP–PBMC SCAP–NKC	2D Direct contact co-culture Direct contact co-culture	SCAPs were significantly more susceptible to killing by IL-2–activated PBMCs. An upregulation of pro-inflammatory cytokines (IFN-γ, IL-6, and TNF-α) by SCAPs when coincubated with PBMCs activated with r-IL2. A total of 70% of SCAPs were lysed when in contact with activated NK cells, while 40% of DPSCs were lysed.
Tatic et al. [153]	SCAP–BV-2 microglial cell	2D Contact-free co-culture in transwell chamber	SCAPs can decrease Tnf and increase the Arg1 expression of LPS-activated microglia when co-cultured in direct contact for 48h.
Liu et al. [154]	SCAP–T cell	2D Direct contact co-culture and contact-free co-culture in transwell chamber	SCAPs have immunomodulatory effects on the Treg conversion in vitro. Cell–cell contact played an important role in the early stage (24 h) of the promotion of SCAPs on Treg conversion, and paracrine effects were also involved in the late stage (72 h).
Kukreti et al. [142]	SCAP–RAW MQ	2D MQ-conditioned media on SCAP	The ability of bioactive engineered nanoparticles to promote stem cell viability, migration, and differentiation potential and reduce inflammation. Release of nitrite and IL-6 was reduced in the presence of SCAPs compared to MQ monoculture.
Kanji et al. [155]	DPSC–RAW MQ	2D Contact-free co-culture in transwell plate	DPSCs inhibit induced osteoclast differentiation of RAW cells and underlying molecular mechanism when co-cultured in a contact-free system.
Croci et al. [156]	DPSC–PBMC	2D Direct contact co-culture PBMC directly on DPSC	Co-cultured with DPSC affected the cytokine profile compared to PBMC monoculture. DPSCs can modulate the production of cytokines deregulated in COVID-19 patients.
Li et al. [128,129]	SCAP–THP-1 MQ	3D Collagen type 1 Cell/matrix on each side with close contact	Highlighted the potential of novel 3D heterogeneous tissue construct for studying cellular response and the role of immune cell–SCAP interactions in periapical inflammation and wound healing. Presented the cell signaling mechanism underlying SCAP–MQ interactions in a novel 3D organoid system simulating inflammation and healing process for a comprehensive understanding of the periapical dynamics of immature tooth for regenerative endodontics.
Anderson et al. [157]	DPSC–RAW MQ DPSC–LoVo gut epithelial cell	2D Contact-free co-culture in transwell plate	Primed (LPS, TNF-α, or IFN-γ) DPSCs altered monocyte polarization toward an immuno-suppressive phenotype (M2), in which monocytes expressed higher levels of IL-4R, IL-6, Arg1, and YM-1 compared to monocytes cultured with control DPSCs. DPSCs induced accelerated wound healing irrespective of priming.
Luo et al. [158]	SCAP–SCAP-derived neuronal cell	3D Collagen type I Cell/matrix on each channel Without direct contact (contact-free)	The central channel contains SCAP-derived neuronal cell spheroids embedded in collagen I hydrogel, and 2 flanking channels contain Oi-SCAP (osteogenic-induced) or SCAP in a conditioned medium. The central channel and the media channels can exchange media and biochemical signals via the gaps between vertical posts. Local microenvironments (with/without neuro-sphere and osteogenic environments) critically regulate the neuro-regenerative potential of SCAP-derived neuronal cell spheroids.

SCAP: stem cell from apical papilla; DPSC: dental pulp stem cell; I-DPSC: dental pulp stem cell derived from symptomatic irreversible pulpitis; PBMC: peripheral blood mononuclear cell; OPC: oligodendrocyte progenitor cell; IDO: indoleamine-pyrrole 2,3-dioxygenase; SCOS: spinal cord organotypic section; NKC: natural killer cell.

**Figure 2 biomolecules-13-00900-f002:**
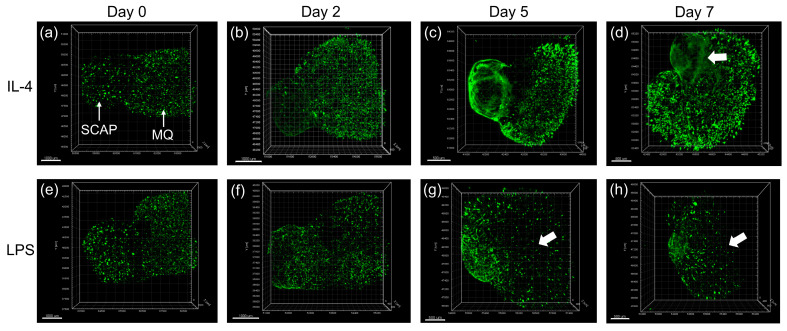
Representative images (**a**–**h**) of formed 3D tissue construct, which is composed of SCAPs and MQs in direct contact at each side. The tissue construct was stained with Calcein-AM from 0 to 7 days in culture. SCAPs and MQs were presented at respective regions. SCAPs formed a distinct cap-shaped structure beginning from day 5 in both inflammatory conditions and were more directionally organized in the IL-4 group, forming a boundary at the junction area (**d**, arrow). The decreased viability of MQs was shown after day 5 (**g**,**h**, arrow). Scale bar = 1000 μm at day 0 and 2; scale bar = 500 μm at day 5 and 7 [128].

**Figure 3 biomolecules-13-00900-f003:**
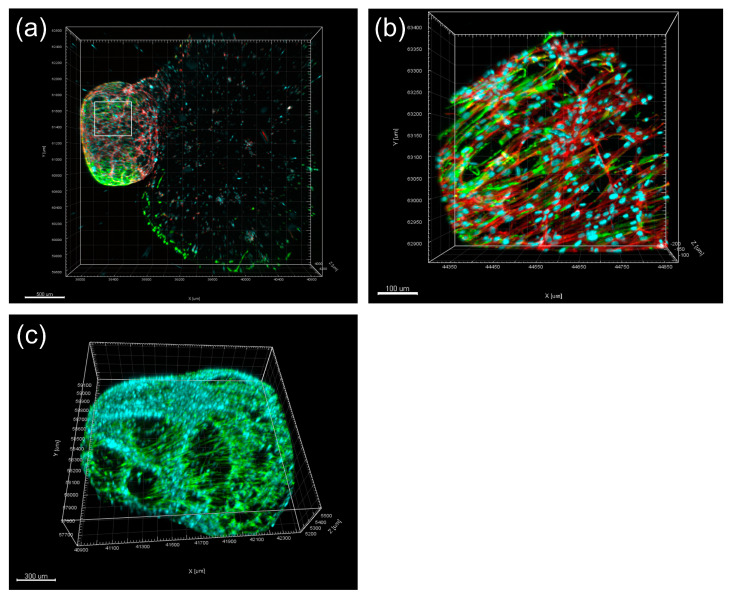
Representative images showed the 3D tissue construct and the morphology of cells. (**a**) Tissue construct formed by the SCAPs (left) and MQs (right) was stained with anti-Vinculin antibody, TRITC-conjugated phalloidin, and DAPI at day 7 of incubation. Scale bar = 500 μm. (**b**) Higher magnification of the SCAPs in 3D tissue construct revealed a unique directional orientation of the spreading of actin-filament towards MQs. Scale bar = 100 μm. (**c**) Distinct apical papilla organoid in the tested 3D tissue construct was observed to organize in an orderly manner into a distinct cap-shaped structure. Scale bar = 300 μm. The tissue construct of SCAPs was stained with anti-dentin sialophosphoprotein (DSPP) antibody followed by FITC-conjugated secondary antibody and DAPI at day 7 of incubation. (Courtesy of Li and Kishen [159].)

## 10. Dentin–Pulp Organoid Developed for Regenerative Endodontics

An organoid is a miniaturized, simplified, and self-organized 3D tissue culture that is derived from stem cells and is designed to imitate much of the complexity of an organ [160]. Organoids associated with the dentin–pulp complex have been developed or fabricated for investigating cell interactions or material testing in the field of regenerative endodontics. In the previous paragraph, the binary cell 3D tissue graft model developed by Li et al. formed a cap-shaped apical papilla organoid. The SCAP–ECM interaction, the dimension of the tissue construct, the environment for free growth, and the expansion of SCAPs in the construct shown in the study replicated the in vivo morphology of the apical papilla (Figure 3b,c). The apical papilla locates at the end of immature roots with a cap-shaped morphology and is composed of rich stem cells that differentiate and form the dentin–pulp complex [55]. The apical papilla organoid revealed a potential model to be used in studying periapical biology, mimicking a disease environment and testing therapeutic strategies in regenerative endodontics. It was applied to study the cell signaling mechanism in inflammatory environments [129]. This model can also be used as an alternative system that bridges the gap between in vitro monoculture-based testing and in vivo animal studies.

Studies including Jeong et al. [161] and Xu et al. [162] developed a dentin-pulp-like organoid using human dental pulp cells (hDPC) with different ECMs. Jeong et al. cultured hDPC with Matrigel for 11 days and expressed the features of both stem cells and differentiated cells (odontoblasts). It was used to test the response to Biodentine, which was the pulp-capping material used in the endodontics [161]. Xu et al. combined hDPC and endothelial cells (EC) with a human-dental-pulp-derived extracellular matrix (hDP-ECM) and cultured them in 3D plates. The organoid fabricated with the co-culture of human hDPC and ECs enhanced in vitro differentiation and mimicked dental pulp tissues, which is crucial for the vascularization of regenerating tissues and bio-fabricated organoids [162]. However, as mentioned in the study, the organoids fabricated in these studies cannot completely establish the structure of the odontoblast layer at the periphery of the pulp and simulate the function of dentin–pulp complex. To mimic the reality of dental pulp tissue, other types of cells or strategies should be considered during dentin–pulp organoid fabrication [162].

## 11. Conclusions

Apical periodontitis and post-treatment healing involve complex interactions between resident stem cells and immune cells. Their crosstalk mechanisms via precise cell-signaling mechanisms can influence the nature and degree of disease progression and post-treatment healing. An ideal model simulating the in vivo scenario is crucial for the comprehensive understanding of cell function and interactions during inflammation and healing, including both repair and regeneration. This will pave the way for drug discovery and effective treatment strategies in regenerative endodontics, while bridging the gap between in vitro 2D cell cultures and in vivo animal/clinical studies. To date, most of the research regarding this field still uses a single type of cells and/or 2D cell cultures set up to investigate cell function and the interaction between cells in control/simulated conditions. This may result in research findings that are not translatable to in vivo cellular functions. A 3D tissue construct that utilizes matrix-based self-assembly techniques enables the formation of a 3D construct with heterogeneous cell types overcoming the limitations of both 2D and 3D cell culture methods. This model can be applied to study the interaction between stem cells and immune cells, as well as study the role of mediators/cytokines in inflammation and healing. These models have the potential to offer a comprehensive understanding of the cell-signaling mechanisms operating in disease and healing. Such 3D models that are well optimized are significant for future development in the field of regenerative endodontics.

## Figures and Tables

**Figure 1 biomolecules-13-00900-f001:**
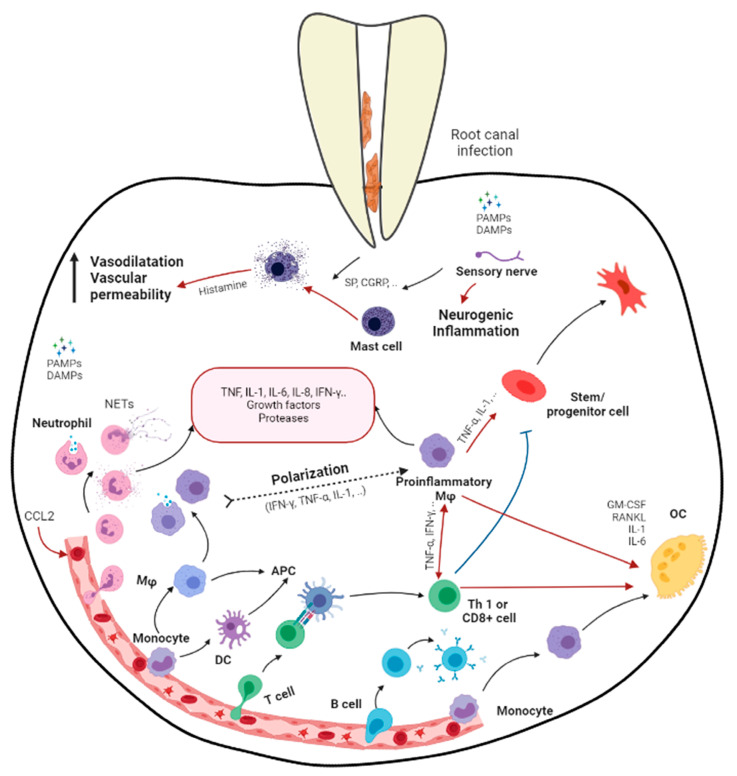
Schematic diagram demonstrating the immune responses in apical periodontitis resulting from the bacterial infection in root canal system (produced by Dr. Hebatullah Hussein). Black arrows indicate a differentiation path or secretion of immune mediators. Black dashed arrows indicate a hypothetical differentiation path. Red arrows indicate induction. Blue arrows indicate inhibition. APC, antigen presenting cell; CGRP, calcitonin gene-related peptide; DAMPs, damage-associated molecular patterns; GM-CSF, granulocyte/monocyte colony-stimulating factor; Mφ, macrophage; OC, osteoclast; PAMPs, pathogen-associated molecular patterns; RANKL, receptor activator of nuclear factor κB ligand; SP, substance P; Treg, regulatory T cell. Created with BioRender.com, accessed on 5 November 2021.

## Data Availability

No new data were created or analyzed in this study. Data sharing is not applicable to this article.

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
