# Peer review of "3D Organoids for Regenerative Endodontics"

_biomolecules, 2023, doi:10.3390/biom13060900_

Round 1

Reviewer 1 Report

The review is very informative. However, due to the extensive content, some aspects of the review are not detailed enough or not new enough, such as 2( Apical Periodontitis)3(Effects of Apical Periodontitis in Immature Root) and  4 ( Challenges in Regenerative Endodontic Procedures).

Suggestion:

1.      The contents of Review 2 and 3 should be compressed and incorporated into the 4 ( Challenges in Regenerative Endodontic Procedures) .

2.      The contents of 5 (Mesenchymal Stem Cells (MSCs) and Stem Cells from Apical Papilla (SCAP) ) and 6 ( Macrophages ) are incorporated into 7 (Mesenchymal Stem Cells - Macrophage Crosstalk).

3.      Focus the review on the content of 8-10.

Author Response

Dear reviewer:

Thank you for the suggestion of the reviewer regarding the content of paragraphs. Authors would like to keep the original sections because of these reasons:

  • The readers of biomolecules journal may not be all dentists/ endodontists. Therefore, separate the 2( Apical Periodontitis), 3(Effects of Apical Periodontitis in Immature Root) and 4 ( Challenges in Regenerative Endodontic Procedures) will provide a full picture to the readers if they would like to understand the relations between disease, the effect of this disease, immature root, and the challenge of current treatment.
  • Section 5 and 6 includes the characteristics and the role of each kind of cells, which is crucial to understand. Separating the paragraphs of each cells between the section of “cross talk” makes it easier for the reader to follow and to avoid a very long section.

Reviewer 2 Report

This manuscript presents an overview of the main challenges to regenerate the pulp-dentin complex and the evolution in cell culture models form 2-D to 3-D, ending with the development of a dental-pulp organoid with interaction between different types of cells. This is an up-to-date review, citing the most relevant papers in this topic and will help researchers to have a comprehensive view of the in vitro models to study endodontic regeneration.

I have no concerns with this manuscript, only detected a typo in Figure 1 – neutrophil.

Author Response

Dear reviewer: 

The figure 1 has been revised accordingly. Typo was corrected. Thank you.

Reviewer 3 Report

I was pleased to review the manuscript “biomolecules-2335316” entitled “3D Organoids for Regenerative Endodontics” for  Biomolecules. This review covers the endodontic disease process, challenges in regeneration, 3D cell culture techniques, and current investigations into regenerative endodontics, including the development of dental-pulp organoids.

In the introduction section, I suggest including a few sentences explaining the concept of 3D organoids briefly.

Please include the following when discussing M1 to M2 macrophage polarization: Dal-Fabbro, R., et al., Next-generation biomaterials for dental pulp tissue immunomodulation. Dent Mater, 2023. 39(4): p. 333-349. doi:10.1016/j.dental.2023.03.013

Overall, I would like to congratulate the authors for writing a thorough and detailed manuscript with a well-developed sequence of topics that addresses the subject of the study.

Author Response

Dear reviewer:

Thank you very much. The reference has been added accordingly (reference 81). 

Reviewer 4 Report

This narative review covers the topic of regenerative endoodntics. Regenerative endodontics has been gaining clinical importance in recent years, especially in the context of treatment of immature teeth with necrotic pulp and apical periodontitis.  

Authors describe relevant cells, molecules, signaling pathways and in vitro research models in inflammation and regeneration processes of pulp and apical periodontium. The article is generally well written. There are some minor issues to be adressed.  

Line 70 form instead of forms….

Ine 70 maybe LPS and lipotheicoic acid as prototypical calsses of PAMP could be mentioned  instead of ''including proteins, carbohydrates, and lipids''

Line 72 induce instead of induces

Line 75 LPS abbreviation should be expalined at the first mention

Line 89 the word ''effec'' is redundant

Line 101 does instead of do

Line 225 Even should be in lower case letter

Line 255 And in lower case letters

Author Response

Dear reviewer:

Thank you very much, all points have been revised accordingly.